# Image quality guided smart rotation improves coverage in microscopy

Jiaye He [1,2,3] & Jan Huisken [1,3]*

Fluorescence microscopy is an essential tool for biological discoveries. There is a constant demand for better spatial resolution across a larger field of view. Although strides have been made to improve the theoretical resolution and speed of the optical instruments, in mesoscopic samples, image quality is still largely limited by the optical properties of the sample. In Selective Plane Illumination Microscopy (SPIM), the achievable optical performance is hampered by optical degradations encountered in both the illumination and detection. Multi-view imaging, either through sample rotation or additional optical paths, is a popular strategy to improve sample coverage. In this work, we introduce a smart rotation workflow that utilizes on-the-fly image analysis to identify the optimal light sheet imaging orientations. The smart rotation workflow outperforms the conventional approach without additional hardware and achieves a better sample coverage using the same number of angles or less and thereby reduces data volume and phototoxicity.

[1] Morgridge Institute for Research, 330 N Orchard St, Madison, WI 53715, USA. [2] Max Planck Institute for Molecular Cell Biology and Genetics, Pfotenhauerstrasse 108, 01307 Dresden, Germany. [3] Department of Integrative Biology, University of Wisconsin Madison, 250 N Mills St., Madison, WI 53706, USA. *email: jhuisken@morgridge.org

Fluorescence microscopy is one of the most important tools in modern biological research. In recent years, there has been an increase in interest amongst biologists in utilizing techniques with lower rates of photodamage, larger field of view and faster volumetric imaging speed. As a result, Selective Plane Illumination Microscopy (SPIM) has become the method of choice for fast three-dimensional imaging of living biological specimens over long time periods. In SPIM, two orthogonally arranged objectives are used for imaging. One objective is used for illumination, where a sheet of light is used to illuminate a thin volume within the sample. The other objective is used for fluorescence signal collection with its focal plane coinciding with the light sheet illumination plane. Compared to conventional point-scanning techniques such as confocal microscopy, the parallelization in illumination and detection of SPIM and the planewise illumination confinement significantly reduces photodamage and increases the speed of imaging.

Many *in vivo* applications require a three-dimensional field-of-view (FOV) of >(500 μm)³. Particularly in such thick samples, aberrations caused by inhomogeneous optical properties can severely degrade the microscope's performance (Fig. 1a). Most common types of aberration include absorption, refraction and scattering. Photon absorption by biological tissue (such as pigments) can cause shadowing effects that become especially apparent in SPIM's orthogonal illumination. Light refraction due to inhomogeneous refractive index can redirect the path of photons, resulting in a range of optical artefacts including defocus and beam steering. Photon scattering by complex biological tissue can broaden the light sheet and decrease the achievable resolution of the optical system (Fig. 1b). The combination of aforementioned effects limits the actual penetration depth of excitation photons. The same degradation also applies to fluorescence photons, resulting in weaker signal and lower than expected detection resolution. Imaging depth is sample and experiment dependent and usually constrained to around 200–300 μm in relatively transparent samples such as zebrafish, not sufficiently deep for large FOV applications such as in toto development imaging.

Multi-view imaging in light sheet fluorescence microscopy is widely used to increase the sample coverage for in toto applications. By taking additional images of the same sample from a different view point and merging these data, the overall image coverage is improved. Additional views can be obtained by rotating the sample with a rotational stage and thereby changing the relative orientation of the sample to the imaging objectives[1,2]. Alternatively, the optical paths can be modified to incorporate additional objectives around the sample. By placing an additional illumination objective opposing the first one, illumination coverage can be improved[3]. Similarly, by adding another detection objective opposing the first one, signal from the sample's far side (relative to the first objective) can be collected with higher efficiency[2,4,5]. Moreover, all objectives can be used for both illumination and detection, giving a total number of 8 views[6]. Multi-objective SPIM is usually faster as multiple cameras can be aligned so that their images are registered inherently without the need for post-processing, while multi-view datasets taken by sample rotation usually require an additional registration step. Otherwise, sample rotation does not substantially increase the overall imaging time with modern, fast stages. Most importantly, sample rotation gives the maximum degree of flexibility in picking the ideal orientation. The two methods have been used in conjunction to efficiently achieve complete coverage of early zebrafish embryos[2].

A priori knowledge of the sample's features that may be optically obstructive and need to be avoided, such as eyes, dense fatty tissue and pigmentation, can assist in selecting better angles for imaging (Fig. 1c). However, in a typical multi-view light sheet experiment, the user defines the angular views manually and often blindly or only based on a qualitative inspection of low resolution images at the beginning of the experiment. The views are usually equally spaced and predefined in number based on the temporal resolution requirement of the experiment and the speed of the camera. Hence, the angle selection process is highly subjective, leading to inconsistent results between imaging experiments. In an attempt to achieve good sample coverage, users often end up acquiring more views than necessary. Unfortunately,

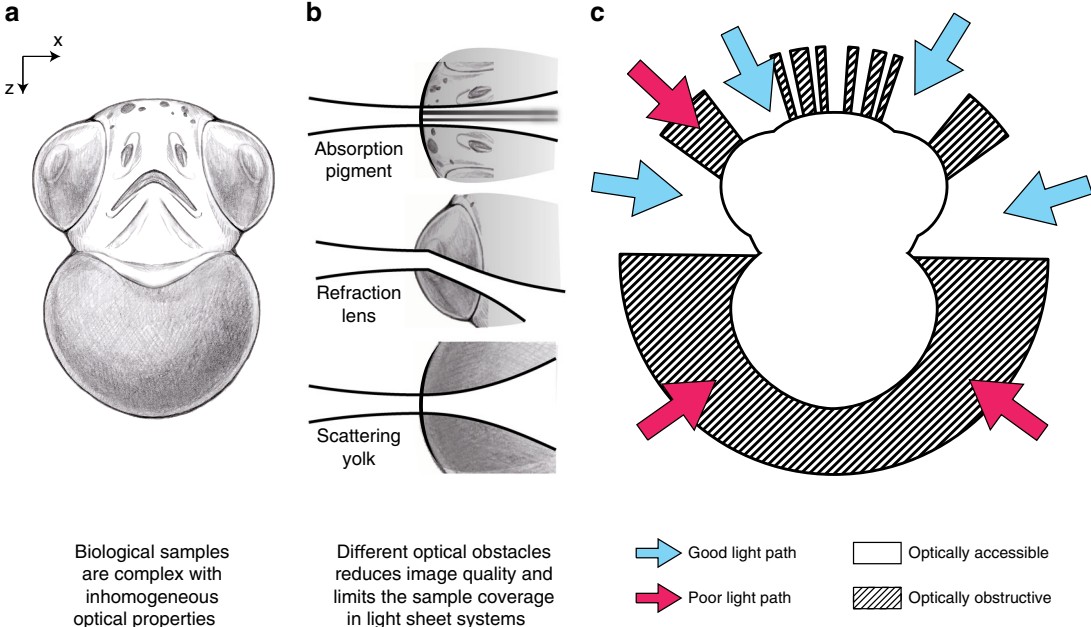

Biological samples are complex with inhomogeneous optical properties

Different optical obstacles reduces image quality and limits the sample coverage in light sheet systems

➡ Good light path

➡ Poor light path

☐ Optically accessible

▨ Optically obstructive

**Fig. 1 Illustrations of common artefacts in optical microscopy. a** Frontal view onto a schematic of a zebrafish embryo along the anterior-posterior axis. **b** Illustration of different optical obstructions in the sample and potential effects on incident excitation light. **c** Illustration showing how the optical obstructions result in predictable quality differences between optical paths.

phototoxicity scales linearly with the number of views and excessive exposure of the sample to laser light can lead to bleaching and severe detriments to sample health[7,8].

In preparation for a time-lapse experiment, users typically cannot account for the natural changes of the sample over time. Here the angular view configuration typically stays the same, although the sample's development affects its optical properties at the same time. Hence, even if the user can manually select a good set of views at one time point, these views may easily become sub-optimal during a time-lapse, yielding unsatisfactory results at later stages. Therefore, a workflow where imaging views are adaptable to sample changes is needed.

In this work, we first evaluate the potential performance of blind multi-view imaging workflows theoretically and in live zebrafish embryos by a method to quantify imaging coverage. We then showcase a new smart rotation workflow that performs on-the-fly image analysis to identify the optimal set of views to maximize sample coverage. We then quantify the sample coverage in a live zebrafish embryo imaged with a multi-view SPIM system[3]. We demonstrate that multi-view datasets taken with the smart rotation workflow have improved sample coverage compared to the conventional approach and that comparable results can be achieved with fewer views.

## Results

**Evaluating sample coverage in a multi-view SPIM system.** In standard 2-lens SPIM (one illumination and one detection objective), the optical coverage is limited by both the illumination and detection path. Only a small angular portion of the sample that is relatively close to both, can be imaged well

(Fig. 2a). Hence, having more views surrounding the sample would offer a more complete coverage (Fig. 2b). Views can be added either through placing additional objectives around the sample or rotating the sample around the vertical axis (Fig. 2c). The aforementioned hypothesis assumes that different angular regions exhibit the same imaging response when illuminated at the same relative angle. Therefore, the angular regions that are far away from both the illumination and detection angle are expected to give worse imaging result. The assumption only holds true if the sample has uniform optical properties and labelling density. However, in biological samples neither the refractive index nor the fluorophore distribution are spatially homogeneous. Hence, blindly applying the assumption to a real biological sample can result in lower information gain than expected when using sub-optimal angles in a multi-view acquisition.

Here we introduce a formulation to evaluate the imaging coverage $C_\alpha$ of an angular region $\alpha$. If the sample coverage is measured in terms of number of foreground voxels, then $C_\alpha$ can be estimated as a von Mises distribution:

$$C_\alpha = \frac{A}{2\pi I_0(\kappa)} e^{\kappa \cos(x - \mu)} \quad (1)$$

Where $x$ is the imaging angle and $\mu$ is the imaging angle where the maximum image coverage for this angular region is achieved. $\kappa$ measures the concentration of the distribution, which reflects the optical accessibility of the angular region. The more concentrated the distribution in $\alpha$ is, the less likely a randomly selected imaging view would provide a good result. The amplitude $A$, encodes the underlying angular fluorophore distribution.

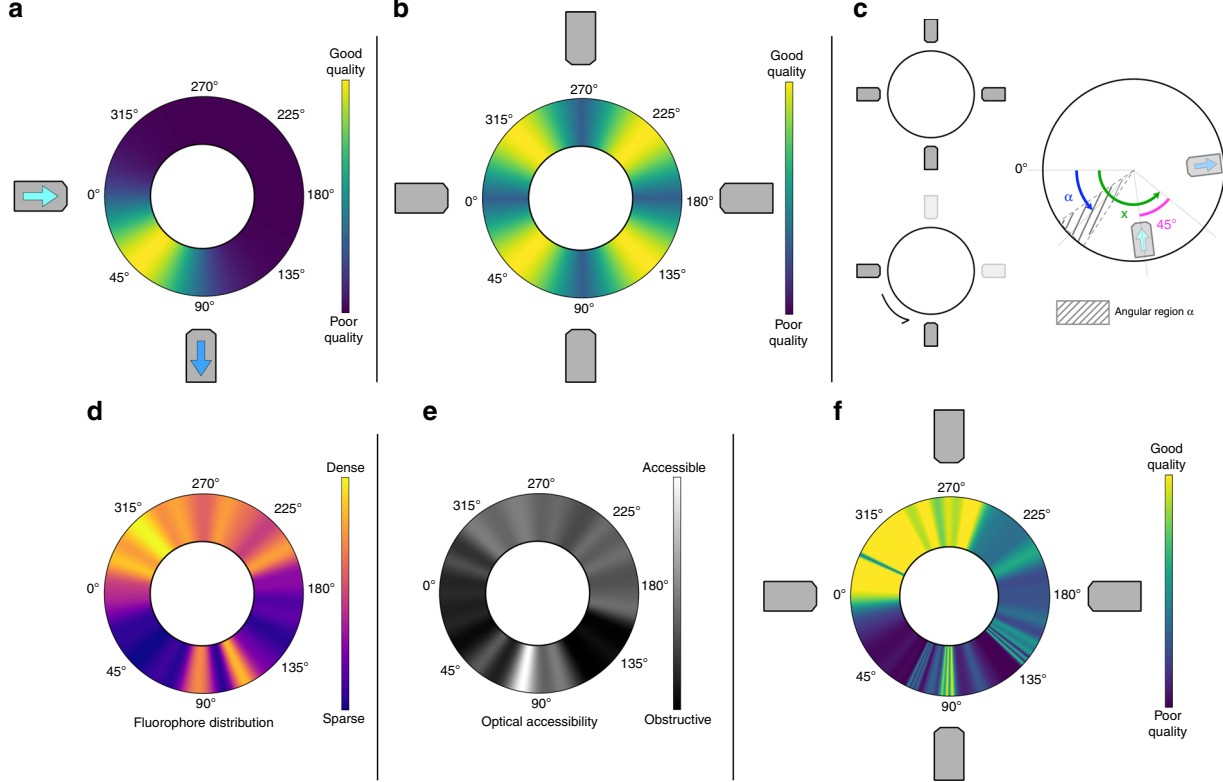

**Fig. 2 Simulation and estimation of sample coverage in a multi-view dataset. a** The simulated sample coverage with a single illumination and a single detection objective. **b** The simulated sample coverage with four views surrounding the sample. **c** The two methods to perform multi-view imaging: Multiple objectives surrounding the sample or multiple views created by rotating the sample. Illustration of variable definitions used in Eq. (1). **d** Estimated angular fluorophore distribution of a 2 day old zebrafish embryo. **e** Estimated optical accessibility of a 2 day old zebrafish embryo. **f** Estimated sample coverage of a real zebrafish embryo measured by number of foreground blocks.

In the conventional multi-view assumption, $A$ and $\kappa$ stay constant whereas $\mu$ is always centered between the illumination and detection paths (Fig. 2a). In this model, increasing the number of (equally spaced) imaging views results in a linear increase in imaging coverage (Fig. 2b, c). In real biological samples, however, angular inhomogeneity in fluorophore distribution (Fig. 2d) and optical accessibility (Fig. 2e) introduce additional complexities in finding the optimal view configuration, resulting in an image with much less content than predicted (Fig. 2f).

To verify the validity of our image coverage measurements, we applied our method to a living animal embryo. Zebrafish (Danio rerio) is a popular model organism in bioimaging applications. SPIM has been applied extensively to zebrafish research in fields such as developmental biology and neurobiology[2,3,9]. We imaged a zebrafish embryo (*Tg(h2afva:h2afva-mCherry)*, 48 hpf), in which all nuclei were labelled, from 24 different angular views for evaluation. To quantify the imaging coverage, we used the discrete cosine transform (DCT) and the Shannon entropy based quality metric to evaluate the information content and image sharpness across the imaging volume[10,11] (See Methods for details). As expected, the angular area of the image that is well-imaged sat between the illumination and the detection objective (Fig. 3a, Supplementary Movie 1) and rotated with the sample. The evaluation metric correctly identified well-imaged areas (Fig. 3c). Data quality degraded significantly as imaging depth increased along both the illumination and detection direction.

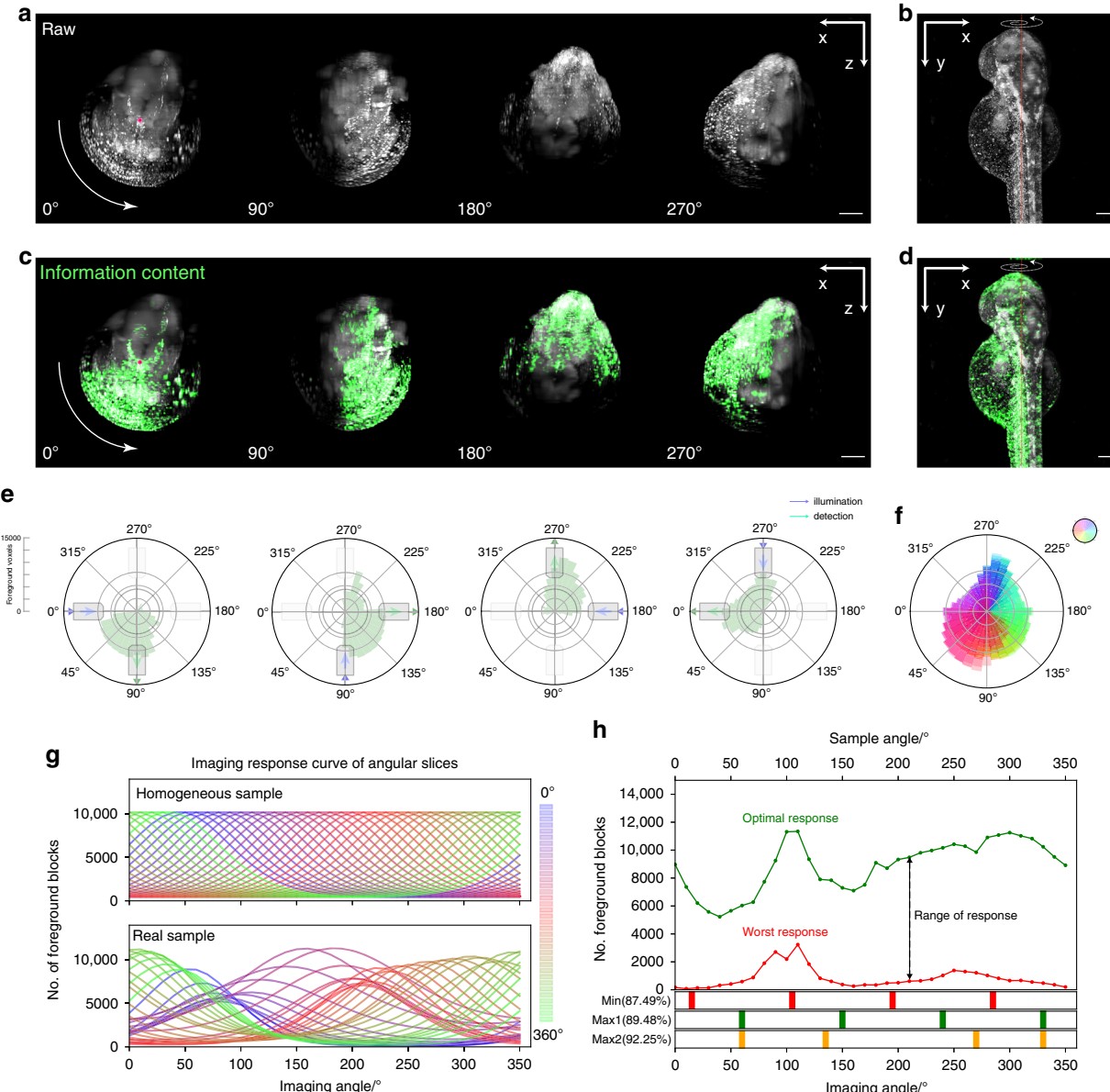

**Fig. 3 Sample coverage evaluation for a zebrafish imaged with 24 views. a** Anterior-posterior (x-z) view of the zebrafish embryo (*Tg(h2afva:h2afva-mCherry)*) at different angles imaged at 48 hpf. Arrow indicates the rotation direction. **b** Dorsal view (x-y) of the zebrafish embryo. **c, d** Information content (green) overlay with raw data (gray). Contrast adjusted with the same threshold between views. **e** Information content summarized by angle of different views measured by $C_\alpha$. **f** Overlaying angular information content of 24 different views, color-coded by angle. **g** Image response curve of angular slices comparison between a completely homogeneous sample and a real zebrafish embryo. Color codes the angular slice. **h** Top: Variation in imaging response for different angular region measured in the number of foreground blocks. Bottom: Comparison of sample coverage of 4 view multi-angle imaging strategies. Min: Minimum coverage given by 4 equally spaced views. Max1: Maximum coverage given by 4 equally spaced views. Max2: Maximum coverage with flexibly view spacings.

However, the fluorophore distribution was highly heterogeneous and optical accessibility spatially varied, resulting in a complex contour of well-imaged area in each view (Fig. 3c–e). The equally spaced views gave varying amount of information and often overlapped in providing good coverage for a specific angular region (Fig. 3f). We then summed the number of foreground voxels by angular region and created a sample response profile (Fig. 3g). The response curves for each angular region were then fitted to Eq. (1). It became evident that different regions had vastly different responses to the imaging angle. The image response curves largely fit the von Mises distribution. There are response profiles that contain more than one peak, meaning that there are alternative imaging angles that provide good imaging results. We opt to only consider the highest peak during fitting but one could build more complex models to include alternative peaks. The coverage varies with the imaging angle. If a blind imaging approach is employed where angles are equally spatially spaced, the overall coverage can be lower than expected due to inhomogeneity. Using a more flexible angular spacing can improve sample coverage (Fig. 3h). If the fluorophore distribution and optical properties are relatively uniform across the sample, the optimal imaging angle for each angular region lies exactly between the illumination and detection objective. In reality, the optimal imaging angle for each angular region lies close to the midpoint between the illumination and detection angle with small variation. However, if we measure the optical accessibility for each angular region as the full width half maximum of each fitted von Mises distribution, we observe a considerable variance (Supplementary Fig. 1).

**Smart rotation workflow to optimize sample coverage.** In addition to evaluating the sample's optical properties, the sample response profile can also be used to predict the overall sample coverage of different sets of views. We optimized the sample coverage estimation and sample imaging response profile generation such that the image analysis can be performed efficiently. We then built a custom microscope control workflow such that the image data can be processed as soon as they are generated and the predicted optimal view combination can be communicated to the microscope control software for reconfiguration (Fig. 4).

To evaluate the imaging coverage accurately, the sample is imaged from $N$ equally spaced angles. The number of angles $N$ needed for evaluation is sample shape and labelling dependent. $N = 24$ angles was generally sufficient to generate a full sample response profile; using fewer angles gave less accurate registration between views, distorting the final coverage estimation. Image stacks are first processed on an Nvidia GPU to perform DCT encoding and Shannon entropy calculation[10,11]. We found CPU processing to be too slow for on-the-fly analysis. The image stacks from different views were registered for direct comparison. Full 3D registration between views is too slow for on-the-fly applications. As there is only one rotational degree of freedom, we chose to summarize the

image in the direction along the rotational axis. The maximum intensity projections of the raw image stacks along the rotational axis are registered with each other using a SIFT based registration method[12]. The transformation is then applied to the minimum intensity projections of entropy image. The entropy image is used to estimate the amount of foreground blocks based on a pre-determined background level. Automated thresholding on the encoded image can also be performed but we found that the entropy measure of information content is not absolute. A slight change in imaging conditions including noise level and pixel exposure time can drastically alter the entropy profile of the encoded image. A predetermined background level can be approximated by the entropy of a blank image, which gives a much more consistent result. Foreground block counts are then summarized by angle with a bin size of 10 degrees to generate the angular information content profile. The information content profile is then fitted to a von Mises distribution (see Eq. (1)) and both optical accessibility $\kappa$, and fluorophore distribution $A$, are calculated for each angular region (Supplementary Movie 2). The performance for each view combination can be estimated as the average coverage percentage of all angular region vs. their estimated optimal coverage. Given the number of views to be imaged for each time point, the combination with the highest average coverage is used for imaging. The combined coverage of views is estimated in an additive manner.

It would also be possible to estimate the optimal view combination in an iterative manner by taking only one view and then adding complimentary views with the most potential information gain. With sufficient prior knowledge of similar samples, iterative estimation can perform comparably. Here we use the 24 equally spaced angles during the estimation step to ensure applicability so that a complete information content map can be obtained for any sample type.

Although SPIM is amongst the fastest imaging modalities to generate full 3D image stacks, generating 24 views with ca. 500 frames each take >2 min. Analyzing the 24 views to generate a complete sample coverage profile takes around 5 min. The entire evaluation step takes roughly 6 min if acquisition and evaluation are performed asynchronously rather than sequentially. In many in toto time-lapse imaging applications, such as zebrafish development studies, image stacks need to be acquired every minute or even faster. Therefore, it is not always feasible to perform a full 24 views evaluation for each time point during a time-lapse. Constantly image with 24 views also exposes the sample to higher photodamage, risking sample health. Instead of performing the full 24-angle evaluation at every timepoint, full evaluation is performed at much longer interval and updates the set of views in use much less frequently.

In cases where subtle optical changes occur over a smaller time interval than the full 24-view evaluation step time interval, an update step can be used. During the update steps, the same analysis is performed on the last acquired views and the overall fluorophore and optical accessibility map can be updated by substituting the corresponding views with the latest views. Since only the newly acquired views need to be processed, update steps evaluation takes

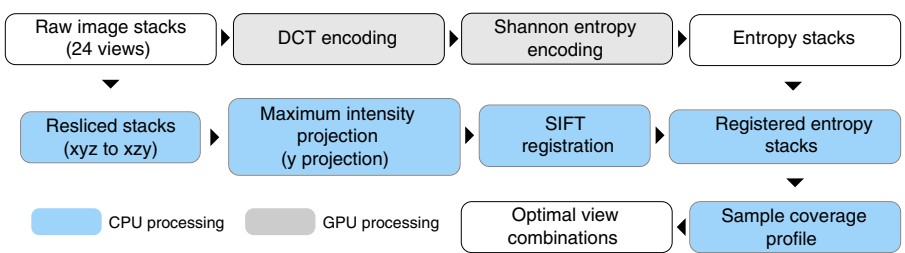

**Fig. 4** The smart rotation workflow evaluation step.

a lot less time and can be run at every time point. In our implementation, the data acquisition runs independently to the image analysis workflow such that data acquisition is not affected in case analysis takes longer than expected. Before the start of each time point, the data acquisition computer communicates with the separate image analysis computer to query for the latest optimal set of views estimated by the workflow.

It is worth noting that down-sampling the image stack can significantly speed up the analysis as the DCT encoding time scales linearly with the number of voxels being processed. Too much down-sampling, however, can result in over-interpolation of the sample coverage profile. A compromise needs to be made to ensure the processing speed can keep up with the data acquisition rate.

**Smart rotation workflow improves in toto imaging coverage**. To verify that our smart rotation workflow improves the sample coverage in light sheet imaging, we imaged a 48-h old zebrafish embryo (*Tg(h2afva:h2afva-mCherry)*). The sample was imaged with 24 equally spaced angles for the evaluation step. The $N = 24$ image stacks from different orientation allowed us to compare the sample coverage performance between the blind multi-view approach that uses equal spacing with a subset of views $n < N$ and the smart rotation workflow with the same number of angles $n$.

To directly compare the sample coverage between the two workflows, multi-view datasets were fused using multi-view reconstruction software[13], which utilizes a bead-based registration. We then performed information content estimation on the fused image stacks. Comparing the sample coverage between fused images from both workflows, the smart rotation workflow was able to consistently outperform the blind approach overall measured in relative coverage percentage (Fig. 5a).

The overall sample coverage increased non-linearly as the number of views $n$ increased (Fig. 5b). There was a diminishing return in additional sample coverage as the number of views $n$ increased. Our smart rotation workflow converges to optimum faster than the blind approach. In typical SPIM experiments, 2–4 views are usually used for time-lapse imaging. With our workflow a similar or better sample coverage can be achieved compared to the blind multi-view approach with 1 less angle used (Fig. 5d). This directly translates to a reduction in the data volume and photo-toxicity in the sample.

To verify that the smart rotation workflow can maintain the sample coverage over a time-lapse experiment better than the blind workflow, we imaged a zebrafish embryo with vascular labeling (*Tg(kdrl:GFP)*) from 48 hpf for 16 h with 4 views. Evaluation steps were performed every 30 min. The 4 angles selected by the smart rotation workflow vary throughout time. The changes in each individual angle remain within 15 degrees (Fig. 5c). The angular image response evolved smoothly (Supplementary Fig. 2 and Supplementary Movie 3). More importantly, during a time-lapse experiment, the sample coverage in a blind multi-view workflow gradually decreases due to the sample's change in optical properties and fluorophore distribution. The sample coverage remained stable with the smart rotation workflow.

In our experience, the optimal configuration does not change significantly when performing time-lapse zebrafish embryo imaging. Therefore, we can run the evaluation step at longer intervals or even omit it for even less phototoxicity.

## Discussion

In this work, we demonstrate the importance of smart multi-view imaging to improving sample coverage in fluorescence microscopy. The widely used blind multi-view imaging workflow does not yield optimal results (for a given number of views) and can result in unnecessary phototoxicity. We formulated a method to evaluate the sample coverage in a multi-view experiment and used this metric to quantify the sample coverage differences between angular views in a living sample with inhomogeneous optical properties and spatial fluorophore distribution. After verifying the metric's performance with data generated from a real multi-view light sheet microscopy dataset, we built the coverage measurement method into a new smart multi-view workflow where optimal view combinations are estimated on-the-fly. We demonstrated that optimal imaging view combinations can be selected during acquisition using our smart rotation workflow. Our workflow not only improves the overall quality of the images captured but also increase the amount of useful information in the data saved. In cases where the evaluation step does not need to run frequently, the total laser exposure time and the risk of phototoxicity are reduced.

Summarizing the information content distribution in 3D into angular slices is a significant simplification in the analysis workflow. It is possible to estimate the global optimum in 3D at pixel resolution since the angular imaging response for all regions is estimated. However, doing so would require a much more stringent 3D registration step, which typically cannot be performed during acquisition. We also tested running the workflow on images down-sampled in z and we saw very small difference on the estimation of the information content distribution. The run-time of the analysis pipeline scales linearly with the number of z-frames and therefore the initial evaluation step can be accelerated by taking images at lower z-sampling rate, reducing the total data volume. Lowering the z sampling also increases the speed of acquisition during the evaluation step. The amount of tolerable down-sampling depends on the intrinsic fluorophore distribution and therefore would need to be tuned for each sample type.

The analysis framework can be further accelerated if the data generated are analyzed in memory before saving to disk to eliminate file writing and reading time. Performing operations on image volumes in memory before saving to disk is usually only possible in home-built solutions. Therefore, we separated the data acquisition part of the workflow and the data analysis section such that users can write an interface layer to incorporate the workflow into their own microscopes. In cases where on-the-fly image analysis is not possible due to hardware limitations, the workflow's evaluation step can still be used as a standalone software to give a better estimation as to what stationary angles should be used in a time-lapse experiment.

Currently the smart rotation workflow is optimized for in toto imaging applications where information-rich voxels can be anywhere within the three-dimensional field of view. In these applications, different views are expected to cover different area of the sample. However, it is also possible to give different weights to different area of interest to optimize the image quality of a specific sub-region. We believe that the information content map itself is an effective representation of the sample that captures both the fluorophore distribution and optical properties. Therefore the map has the potential to be used as a template to estimate the optimal imaging condition for new experiments where similar samples have been imaged before.

To fully utilize the power of multi-view imaging, post-acquisition data fusion is required[1,14]. In our experience, a weighted average method of image fusion can actually decrease the overall image quality compared to the individual views: The well-imaged part of one view may be corrupted by another view with poor image quality in this region. In the future, we hope to extend the information content map to estimate how a

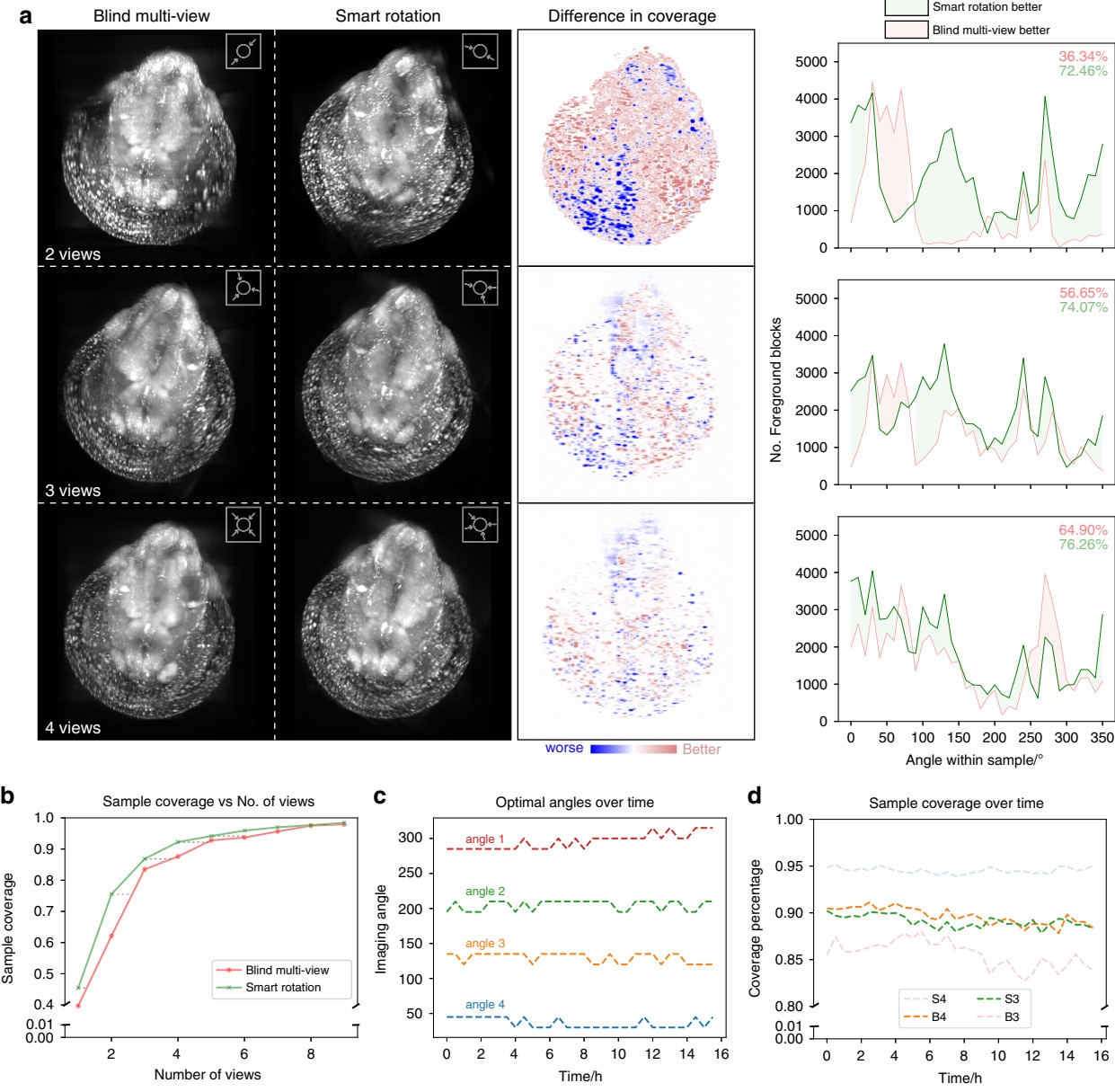

**Fig. 5 Performance comparison between blind multi-view imaging and the smart rotation workflow. a** Comparison between the fused images from a blind multi-view approach and our smart rotation workflow. The difference image is calculated as the difference in information content between the smart rotation generated image and the blind multi-view generated image. Number denotes relative image coverage. **b** Sample coverage measured as a percentage of the maximum possible against the number of views used. **c** Angles selected by the smart rotation evaluation during a 16 h time-lapse of a zebrafish embryo (*Tg(kdrl:GFP)*). **d** Sample coverage percentage comparison between the smart rotation workflows and blind multi-view workflow over the 16 h time-lapse. S denotes smart rotation workflow and B denotes blind multi-view workflow and number denotes and number of view used.

specific view would affect the eventual fused result and adjust the weighting accordingly. Adaptive cropping of the data may also yield better fusion results and will be more robust than manual cropping, which is sometimes done.

This workflow illustrates a first step towards a smart, content-aware microscope[15]. The workflow can be adapted to different imaging modalities that utilize rotational multi-view. We envision that the principle of image analysis guided microscopy can be extended to more complex operations. In this work, only the rotational degree of freedom is controlled by image analysis. Eventually a smart microscope would go beyond a simple observation tool and become an automated exploratory instrument where only relevant and not redundant data are recorded,

yielding the maximally achievable resolution across the entire sample and across time.

## Methods

**Information content assessment**. Information content assessment is performed using a combination of discrete cosine transform (DCT-II) and Shannon entropy similar to pervious approach to measure image quality[11]. Image patches are transformed into the cosine frequency domain (Eq. 2). The spectral entropy is then used to calculate the information content of the patch (Eq. 3):

$$F_{dct}(u,v) = \frac{2}{N} \sum_{i=0}^{N-1} \sum_{j=0}^{N-1} \delta(i)\delta(j) \cos\left[\frac{\pi u}{2N}(2i+1)\right] \cos\left[\frac{\pi v}{2N}(2j+1)\right] I(i,j) \quad (2)$$

where,

$$\delta(t) = \begin{cases} \frac{1}{\sqrt{2}} \ \forall \ t = 0 \\ 1 \ \forall \ t \neq 0 \end{cases}$$

$$S_{shannon} = -\sum_{i=1}^{n}\sum_{j=1}^{n} p_{i,j} \ln p_{i,j} \qquad (3)$$

where,

$$p_{i,j} = \frac{P_{i,j}}{\sum_{i,j} P_{i,j}}$$

$$P_{i,j} = \frac{F_{dct}(i,j)^2}{N^2}$$

Where the original image intensity at pixel $i$, $j$ is $I(i, j)$. $N$ is the size of the patch. $F_{dct}$ is the discrete cosine transform of the 2D image patch. $S_{shannon}$ is the Shannon entropy of the transformed image.

**Zebrafish samples and embedding**. Zebrafish (Danio rerio) embryos and adults are kept according to established protocols[16]. Zebrafish husbandry and maintenance were conducted according to protocols approved by the UW-Madison Institutional Animal Care and Use Committee (IACUC).

Transgenic lines *Tg(h2afva:h2afva-mCherry)* and *Tg(kdrl:GFP)* are used for experiments. For imaging experiment, embryos are collected at 48 h post fertilization and kept at 28.5 degrees Celsius. In SPIM experiments, sample is embedded in FEP tubes (0.8 mm inner diameter, 1.2 outer diameter, Bola) with 0.8% low melting point agarose (Sigma) made with E3 containing 200 mg per l Tricaine (Sigma). The sample holding FEP tube is then placed in a E3 filled chamber also containing 200 mg per l Tricaine. To demonstrate fusion result, fluorescent beads (Fluoresbrite Plain YG 0.5 micron microspheres, Polyscience) are also mixed in the embedding agarose gel at a 1:10000 ratio[13].

**Multi-view light sheet microscope**. A custom built SPIM is used for imaging. The optical configuration is similar to previously published system[3]. The divergent output from a fiber coupled laser engine (Toptica iChrome MLE) is collimated and then expanded. The expanded beam is reshaped with a cylindrical lens and projected on to the back of the illumination objective (Olympus UMPLFLN 10× W). The beam is pivoted with a resonant mirror (SC-10, EOPC) to reduce the shadowing effect caused by the sample. An orthogonally arranged detection lens (Olympus UMPLFLN 10× W) is used to collect fluorescence from the illuminated plane. Collected signal is then filtered and imaged onto an sCMOS camera (Andor Zyla 4.2 Plus). Sample containing FEP tube is placed on a translational stage assembly (M-111.1DG, PI) and a rotational stage (U-651, PI). The sample holder is modified so that the sample can be translated in the plane perpendicular to the rotation axis. The axis of rotation is aligned to the center of the image plane according to previously published protocol for OPT alignment[17].

**Microscope control and Smart rotation workflow**. The multi-view SPIM is controlled by a custom program written in LabVIEW (National Instruments). The smart rotation analysis software is written in both Java and Python. Data generated by the microscope is streamed directly to a centralized fileserver (F800, Dell EMC Isilon). A network attached analysis machine (Dell PowerEdge R730) with a CUDA-enabled GPU (Nvidia Quadro P5000) runs the analysis server which accepts analysis request from the LabVIEW control software. The majority of the analysis software is based on existing Fiji plugins[18]. The GPU code is custom written in C++ and bind to java using the JCUDA[19]. The microscope control machine and the analysis machine communicate via the TCP/IP protocol.

Currently, the software hosted on the github repository (https://github.com/henryhetired/smartrotationjava) can be run as a command line software to evaluate the sample coverage of a multi-view SPIM dataset and return the optimal angle combination given the number of angles needed. During imaging, the software runs on a separate computer to the microscope control computer. The software listens to commands sent by the control computer via TCP/IP to perform the necessary analysis and return the results. The capture machine then reconfigures for the subsequent imaging steps. The smart rotation workflow can be integrated into any SPIM where the users have access to the underlying control software. The detailed command communication structure can be found on the repository. Even if it is not possible to integrate the smart rotation workflow into the image acquisition process, users can acquire a 24-view dataset and run the sample coverage estimation manually. This allows the user to find an optimal configuration at the beginning of the experiment, which is often still better than blind angle selection. The user can also use the command line tool to generate a figure similar to Fig. 5b to see how the sample coverage improves as the number of views increases. This allows the user to determine the appropriate number of views to use during imaging. For detailed instructions on the command line interface, see the github repository.

**Reporting summary**. Further information on research design is available in the Nature Research Reporting Summary linked to this article.

## Data availability
The data generated and analyzed in this study are available from the corresponding author upon reasonable request.

## Code availability
The code used to perform the image analysis task is available at https://github.com/henryhetired/smartrotationjava.

The LabView microscope control code and the raw image data used in the study are available upon request.

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

## Acknowledgements
We thank A. Graves for helping with zebrafish protocols and husbandry. J. Whisenant for the scientific illustration of Fig. 1a–c, N. Scherf for the fruitful discussion and suggestion, N. Gritti and R. Power for manuscript discussion and the Morgridge IT team for the infrastructure support. Work done in the Huisken lab was funded by the Morgridge Institute for Research and the Max Planck Society.

## Author contributions

Jiaye He and Jan Huisken conceived the idea. Jiaye He developed the microscope instrumentation, wrote the software for image analysis and performed the experiment. Jan Huisken supervised the imaging experiment. Jiaye He and Jan Huisken wrote the manuscript.

## Competing interests

The authors declare no competing interests.
