## [Peer Review File · Nature Communications]

Reviewers' comments:

Reviewer #1 (Remarks to the Author):

Jiaye He and Huisken present a beautifully executed concept of smart microscopy. In light sheet imaging, signal attenuation along the illumination and detection axis hampers complete imaging coverage of large specimen. Multi-view acquisition to some degree ameliorates the problem, however until now, angles of a multi-view acquisition have been chosen arbitrarily. He and Huisken show convincingly that we can do much better than that by first evaluating signal degradation in a given specimen and subsequently selecting a set of angles that maximizes coverage of the specimen. The presented paper does a great job at convincing the reader that this approach is better and that it is in principle possible to implement it in a custom microscopy set-up. The approach to evaluate sample coverage and the optimisation scheme is probably one of many, but it is demonstrated that on the presented sample it works. There is a lot of engineering behind the approach both on the microscopy hardware and software side. The manuscript is very well written.

I have few suggestions to improve the manuscript:

1) As far as I understood, the number of angles that the software is meant to select from the more detailed scan (many more angles, here 24) is fixed and constant across the timelapse. Did the authors consider extending the optimisation scheme to find the optimal set (number) of angles to cover a given specimen? Such optimisation may become ill-posed, however it could be constrained by the desired frame rate of the acquisition.

2) It is somewhat unclear what would it take to reproduce this, so to say, at home. One needs to have a multi-view SPIM microscope that is being controlled by a certain software (in case of the authors it is a proprietary LabView code). There is some source code provided, however it is unclear what parts of the pipeline it performs. There is no explanation, how this software could be connected to one's favourite microscope control package. It would have been nice to implement this concept for the OpenSPIM platform that the authors used to support. At this point, it is beyond the scope of the study. However, the authors should provide an honest discussion of the obstacles towards reproduction of their prototype.

3) Phototoxicity of SPIM (or lack thereof) is discussed at number of places in the manuscript. I think it would be appropriate to cite the recent reviews on the subject by Icha et al. and Laissue et al.

4) Fusion approaches are critical to this study, they have been published and should be cited.

5) page seven, I would insert comma after "close to both", I got stuck on that sentence.

6) As I mentioned above, the paper reads very well. It appears a little bit wordy, especially in the introduction section, where some parts are redundant with results. It could be edited down.

Finally, I have a small disclosure to make. I had this idea many years ago and after I mentioned it to Zeiss colleagues, they wrote a patent on it. It has absolutely no bearing on the presented work. I never did anything about it and as far as I know neither did Zeiss. I am very happy to see it realised.

Pavel Tomancak

Reviewer #2 (Remarks to the Author):

The authors demonstrate a new workflow to perform multi-view imaging using the light-sheet fluorescent microscopy. The authors sought to select a combination of optimal angles to efficiently reconstruct the sample information in comparison with the conventional multi-view methods. Twenty-four equally spaced angles were applied to assess photon penetration and fluorophore distribution. The strengths of this manuscript reside in the demonstration of enhanced image

quality with reduced photo-toxicity. The innovation appears to be incremental as compared with the existing methods.

Comments:

1. The proposed method outperformed the blind multi-view in < 4 views; however, the results of the blind multi-view can become similar to the new method by increasing the viewing angles. In general, the blind multi-view is sufficiently robust with respect to the variance of sample optic character, and the users are able to perform > 4 views to improve the image quality. For these reasons, the new method seems not to address the critical issue in multi-view imaging. In light of the bead-based registration, the computational resources increase exponentially in relation to the increasing number of views and computing cost.

2. Pre-conducting 24 views to evaluate the sample's optical characters would support the reduction in photo-toxicity, and periodical assessment would track the changes to optimize the response. However, the time interval of 6 minutes during the evaluation process may reduce the temporal resolution, and the additional evaluating step may increase the sample exposure to the laser illumination.

3. The authors demonstrate an optional updating method by using the latest view to replace the corresponding view in the overall fluorophore and optical accessibility map. This option may be insufficient to update the character of optical change with a limited number of views.

4. To improve the efficiency of the method, the authors may further consider adaptively predicting the next appropriate imaging angle by using the previous information rather than using equally spaced 24 views.

5. The authors propose that the imaging coverage would fit the von Mises distribution without any specific background and theoretical analysis. Please articulate the relationship between imaging coverage and von Mises distribution. Is this a genuine distribution for imaging coverage, or a fitting curve subject to this type of distribution? The optimal response curve seems not to support the conclusion.

6. Optimization of current multi-view methods seems incremental and redundant for zebrafish as evidenced by the backgrounds and figures in both the abstract and introduction. Unclear is how this new method has the capacity to unravel new applications from zebrafish otherwise challenging with the existing imaging modalities.

Reviewer #1

Jiaye He and Huisken present a beautifully executed concept of smart microscopy. In light sheet imaging, signal attenuation along the illumination and detection axis hampers complete imaging coverage of large specimen. Multi-view acquisition to some degree ameliorates the problem, however until now, angles of a multi-view acquisition have been chosen arbitrarily. He and Huisken show convincingly that we can do much better than that by first evaluating signal degradation in a given specimen and subsequently selecting a set of angles that maximizes coverage of the specimen. The presented paper does a great job at convincing the reader that this approach is better and that it is in principle possible to implement it in a custom microscopy set-up. The approach to evaluate sample coverage and the optimisation scheme is probably one of many, but it is demonstrated that on the presented sample it works. There is a lot of engineering behind the approach both on the microscopy hardware and software side. The manuscript is very well written.

I have few suggestions to improve the manuscript:

1) As far as I understood, the number of angles that the software is meant to select from the more detailed scan (many more angles, here 24) is fixed and constant across the timelapse. Did the authors consider extending the optimisation scheme to find the optimal set (number) of angles to cover a given specimen? Such optimisation may become ill-posed, however it could be constrained by the desired frame rate of the acquisition.

Thanks for the suggestion. We have now implemented a feature such that when the user runs a full 24 view evaluation, a plot similar to Figure 5.b is generated to inform the user how the sample coverage changes as the number of views increases. This allows the user to determine a reasonable number of views to use.

We have included the following sentences explaining this new feature.

“The user can also use the command line tool to generate a figure similar to Figure 5.b to see how the sample coverage improves as the number of views increases. This allows the user to determine the appropriate number of views to use during imaging.”
(see page 24, line 6-9)

2) It is somewhat unclear what would it take to reproduce this, so to say, at home. One needs to have a multi-view SPIM microscope that is being controlled by a certain software (in case of the authors it is a proprietary LabView code). There is some source code provided, however it is unclear what parts of the pipeline it performs. There is no explanation, how this software could be connected to one's favourite microscope control package. It would have been nice to implement this concept for the OpenSPIM platform that the authors used to support. At this point, it is beyond the scope of the study. However, the authors should provide an honest discussion of the obstacles towards reproduction of their prototype.

Thanks for the great comment! The evaluation and selection of the optimal angles is done independent of the microscope control and therefore not dependent on its implementation. We have included a more detailed description on how other people can implement the workflow in their SPIM:

“Currently, the software hosted on the github repository can be run as a command line software to evaluate the sample coverage of a multi-view SPIM dataset and return the optimal angle combination given the number of angles needed. During imaging, the software runs on a separate computer to the microscope control computer. The software listens to commands sent by the control computer via TCP/IP to perform the necessary analysis and return the results. The capture machine then reconfigures for the subsequent imaging steps. The smart rotation workflow can be integrated into any SPIM where the users have access to the underlying control software. The detailed command communication structure can be found on the repository. Even if it is not possible to integrate the smart rotation workflow into the image acquisition process, users can acquire a 24-view dataset and run the sample coverage estimation manually. This allows the user to find an optimal configuration at the beginning of the experiment, which is often still better than blind angle selection. The user can also use the command line tool to generate a figure similar to Figure 5.b to see how the sample coverage improves as the number of views increases. This allows the user to determine the appropriate number of views to use during imaging.”
(see page 23, line 16 – page 24, line 9)

3) Phototoxicity of SPIM (or lack thereof) is discussed at number of places in the manuscript. I think it would be appropriate to cite the recent reviews on the subject by Icha et al. and Laissue et al.

Thanks. We have included to citations for Icha et al. and Laissue et al. in the manuscript (new ref. 7).

4) Fusion approaches are critical to this study, they have been published and should be cited.

That is absolutely correct. We have included citations (refs. 1, 14)

5) page seven, I would insert comma after "close to both", I got stuck on that sentence.

Thanks.

6) As I mentioned above, the paper reads very well. It appears a little bit wordy, especially in the introduction section, where some parts are redundant with results. It could be edited down.

Thanks. We have tried to shorten the manuscript, in particular the introduction.

Reviewer #2

The authors demonstrate a new workflow to perform multi-view imaging using the light-sheet fluorescent microscopy. The authors sought to select a combination of optimal angles to efficiently reconstruct the sample information in comparison with the conventional multi-view methods. Twenty-four equally spaced angles were applied to assess photon penetration and fluorophore distribution. The strengths of this manuscript reside in the demonstration of enhanced image quality with reduced photo-toxicity. The innovation appears to be incremental as compared with the existing methods.

Comments:

1. The proposed method outperformed the blind multi-view in < 4 views; however, the results of the blind multi-view can become similar to the new method by increasing the viewing angles. In general, the blind multi-view is sufficiently robust with respect to the variance of sample optic character, and the users are able to perform > 4 views to improve the image quality. For these reasons, the new method seems not to address the critical issue in multi-view imaging. In light of the bead-based registration, the computational resources increase exponentially in relation to the increasing number of views and computing cost.

As the reviewer correctly points out, one of the challenges in multi-view SPIM imaging is the overwhelming amount of data generated. We have demonstrated that in some cases, we are able to achieve a similar sample coverage with less imaging views using the smart rotation workflow (Figure 5 (d)). Therefore, we can reduce the number of images that need to be generated, alleviating some data processing stress. We agree that it is much easier to image the sample with as many views as possible given the time constraint. However, the amount of photo-toxicity increases linearly with the number of views used. It is beneficial for the sample health to reduce the number of views at each timepoint.

(see page 5-6 for discussion)

2. Pre-conducting 24 views to evaluate the sample's optical characters would support the reduction in photo-toxicity, and periodical assessment would track the changes to optimize the response. However, the time interval of 6 minutes during the evaluation process may reduce the temporal resolution, and the additional evaluating step may increase the sample exposure to the laser illumination.

Thank you. It is true that the smart rotation workflow with periodic re-assessment is not suitable for some applications requiring high temporal resolution. The amount of time needed to perform the evaluation step is the limiting factor of the speed of the workflow. We mentioned that there are ways to improve the speed of the evaluation step including using fewer z-planes per stack. In terms of photo-toxicity, we agree that constantly performing the evaluation step would significantly increase the photo-toxicity, which is why we are performing the evaluation step at longer time intervals. In our experience, the optimal configuration does not change significantly over the course of a zebrafish embryo time-lapse. Therefore, we can run the evaluation step at much longer interval or even omit it. It is possible that for certain samples, constant re-evaluation is needed to maintain optimality. However, in the samples we have tested, re-evaluation every hour or longer is usually sufficient.

We have added the following for clarification:

“In our experience, the optimal configuration does not change significantly when performing time-lapse zebrafish embryo imaging. Therefore, we can run the evaluation step at longer intervals or even omit it for even less phototoxicity.”

(see page 18, line 10-12)

3. The authors demonstrate an optional updating method by using the latest view to replace the corresponding view in the overall fluorophore and optical accessibility map. This option may be insufficient to update the character of optical change with a limited number of views.

Thank you for the comment. Using only a few angles to update the overall optical accessibility map is a compromise between using only the previous step and running the full evaluation at every time point. If the imaging speed is sufficient to resolve the dynamics in the sample, it is reasonable to assume that the optical accessibility map either does not change much or evolves smoothly. Therefore the update steps should be sufficient in correcting minor drifts in the optimal angles.

4. To improve the efficiency of the method, the authors may further consider adaptively predicting the next appropriate imaging angle by using the previous information rather than using equally spaced 24 views.

Thank you so much for the suggestion. We have considered ways to trigger the evaluation step on-the-fly rather than having static evaluation intervals. In theory one can monitor the information content provided by each imaging view. If any one view's information content drops below a certain pre-defined percentile of previous time points, a new evaluation step is triggered. However, it would be difficult to deal with situations where the sample is rapidly developing and the information content profile is constantly varying due to biological changes.

5. The authors propose that the imaging coverage would fit the von Mises distribution without any specific background and theoretical analysis. Please articulate the relationship between imaging coverage and von Mises distribution. Is this a genuine distribution for imaging coverage, or a fitting curve subject to this type of distribution? The optimal response curve seems not to support the conclusion.

We are happy to give more details on our formulations. Our assumption is that, for each angular region, there exists an optimal imaging angle to give the maximum amount of information. The further the imaging angle deviates away from the optimal angle, the less information is contained in the image. Therefore, we use a von Mises distribution to model this process as von Mises distribution can be considered a circular gaussian distribution. When we tested the formulation on real biological data, we found that the imaging response mostly agree with the formulation (see supplementary vid 2). We notice that the fitted curve is not perfect. Certain angular region may have been optically accessible from multiple angles and therefore the response curve would be a combination of multiple von Mises node. In the future, one could build more complex models to resolve the issue. We still think that the von Mises distribution serves our purpose well and has the necessary characteristics. We have included the following in the manuscript to clarify this:

"The image response curves largely fit the von Mises distribution. There are response profiles that contain more than one peak, meaning that there are alternative imaging angles that provide good imaging results. We opt to only consider the highest peak during fitting but one could build more complex models to include alternative peaks." (see page 11, line 23 and following)

6. Optimization of current multi-view methods seems incremental and redundant for zebrafish as evidenced by the backgrounds and figures in both the abstract and introduction. Unclear is how this new method has the capacity to unravel new applications from zebrafish otherwise challenging with the existing imaging modalities.

The scope of our manuscript is to describe our novel adaptive rotation method and demonstrate its performance in living zebrafish embryos. At this moment our intention is to showcase the performance and inspire other scientists to implement this or similar approaches to make light sheet microscopy more efficient and more suitable for more challenging samples. We believe that we are making an important contribution to advance light sheet microscopy and to achieve better penetration and coverage in optically challenging biological samples, which will open more possibilities in studying the anatomy and development of those samples. The advances may seem incremental, but we and others have shown in the past how innovations in microscopy and image analysis can lead to new fundamental insights in biology.

Reviewer #1 (Remarks to the Author):

Great revision. Looking forward to see this used, especially for complex shaped specimen.

Pavel Tomancak

Reviewer #2 (Remarks to the Author):

The investigators provided a solid rotational strategy to mitigate photo-toxicity. They eloquently addressed a pipeline to quantify the sample coverage and to boost multi-view imaging in this re-submission.

Remain unclear is whether the pipeline would have the capacity to demonstrate how this smart rotation is capable of reducing photo-toxicity for time-lapse imaging or to enhance optical access for deep photon penetration that are otherwise challenging with the existing approaches.